# Equity in practice: Assigning competence to shape STEM student participation

**Daniel Lee Reinholz**[1]*, **Mariah Gabriella Moschetti**[1], **Jan Tracy Camacho**[1], **Eva Fuentes-Lopez**[1], **Charles Wilkes II**[1], **Niral Shah**[2]

**1** Department of Mathematics and Statistics, San Diego State University, San Diego, CA, United States of America, **2** Learning Sciences and Human Development, University of Washington, Seattle, WA, United States of America

* daniel.reinholz@sdsu.edu

**Data Availability Statement:** All relevant data are within the manuscript and its Supporting Information Files.

**Funding:** This material is based upon work supported by the National Science Foundation

## Abstract

Improving equity in undergraduate STEM is a national imperative. Although there is a rapidly growing body of research in this area, there is still a need to generate empirical evidence for equitable teaching techniques. We ground our work in Complex Instruction, an extensively researched pedagogical approach based on sociological theories and the malleability of status. This approach has been applied primarily in K-12 classrooms. In this manuscript, we explore the application of one strategy from Complex Instruction—assigning competence—to undergraduate STEM classrooms. We provide an analysis of three instructors' implementation of assigning competence and track the impact on student participation. This work makes a unique contribution to the field, as the first study that directly documents changes in student participation resulting from assigning competence in undergraduate STEM.

## Introduction

Improving equity in STEM education is a national imperative [1]. Especially over the past few years, in response to multiple pandemics–COVID-19 and police violence against Black Americans–conversations around equity and racial justice have come to the fore on college campuses. Consequently, campuses have begun to offer a wide variety of workshops on bias, microaggressions, inclusive classrooms, and so forth. While those approaches may lead to greater awareness, in isolation, they rarely impact the implementation of equitable *practices* [cf. 2, 3]. One obstacle is that the goals of "equity" and "inclusion" are nebulous, and when they are operationalized, it is often through outcome data (e.g., GPAs, course passage rates). These data come too late to be actionable (i.e., if a student has already failed the course, it is too late to prevent them from failing). Overall, this has created barriers to the uptake and study of equitable teaching strategies. From an educational research standpoint, there is still limited empirical evidence documenting the impact of most inclusive teaching strategies [4].

Perhaps the most widely studied set of techniques—and one of the few exceptions to the above statement—is Complex Instruction [5]. Complex Instruction is a set of practices grounded in sociological theory that has developed over five decades. This program of research began with laboratory studies, and later transitioned to classroom studies. Despite the

([https://www.nsf.gov/](https://www.nsf.gov/)) under Grant No. 1943146 awarded to DR. The funders had no role in study design, data collection and analysis, decision to publish, or preparation of the manuscript.

**Competing interests:** The authors have declared that no competing interests exist.

impressive body of work, no program of research is ever truly complete, and some important questions remain unanswered. For example, while many studies have shown the positive impact of Complex Instruction on learning outcomes overall [6, 7], there is less work that tracks the impact of these techniques on *individual students* who are perceived as low-status and from marginalized groups. This is the gap filled by the current work. In this paper, we focus on one specific technique from Complex Instruction, called *assigning competence*. Assigning competence is a teaching strategy that focuses on noticing the contributions of students perceived as low status and positioning those contributions publicly in a positive way to elevate the status of those students.

The data from this manuscript come from a larger, multi-phase study that engaged university STEM faculty in sustained professional learning for one or more semesters [8]. Participants in the study were provided with data analytics describing patterns of student participation in their classrooms, and they engaged in iterative cycles of observation, reflection, and revision to their teaching practices. In response to the equity data, instructors adopted a variety of instructional practices to mitigate racial and gender inequities in their classrooms. Given the extensive scope of the larger multi-phase study, we have reported on portions of the work in different venues. In this paper, we focus on a subset of the faculty members that attended specifically to assigning competence as one of their instructional strategies. As such, this dataset gives us a unique opportunity to track the impact of assigning competence on specific students within university STEM classrooms. Student participation was tracked using the EQUIP observation tool [9], which offered a rigorous methodological approach to understanding and tracking changes in student participation (an indicator of status). This allowed us to answer the following research questions:

1. *How did instructors leverage assigning competence as an instructional strategy to mitigate racial and gender inequities in their classroom participation*?

2. *How did the use of assigning competence impact classroom participation patterns*?

The manuscript is organized as follows. We begin with an overview of Complex Instruction and prior studies of assigning competence. This sets the stage for describing the context and methods of the current study. In the results section, we first provide an overview of instructors' use of assigning competence and code all instances of assigning competence according to a typology of the strategy [10]. Next, we track the impact on student participation patterns over time. Finally, we provide qualitative vignettes to illustrate the impact on changes in participation over time.

## Background

### Status and status interventions

The concept of status has been developed over many years as a part of sociological theory [6]. Broadly, status can be understood as a student's social standing in the classroom. High-status students are seen as more capable of contributing valuable ideas, whereas low-status students are more likely to be ignored. Status is indicated through social interactions and depends on how an individual student is perceived by their teacher, their peers and by themselves. Status is often related to other student characteristics such as academic achievement, popularity, language proficiency, and social attraction. Status is particularly salient in STEM settings, where some students have been positioned as innately good at these subjects [e.g., 11], and others are positioned as not belonging [12]. Notably, these forms of positioning tend to fall along student

social marker identities such as race, gender, and disability [13–15]. As such, perceived status tends to align with existing social marker inequities in society.

The early work around status began in laboratory settings. In one study, the close connection between status and student participation was documented [16]. In this study, when researchers tracked student contributions to conversations, most of the contributions came from students perceived as higher status. A parallel laboratory study documented the malleability of status through targeted interventions. This study focused on changing the task design and framing to shift the status of Black students as compared to White students in small groups [17]. The intervention had a measurable impact by increasing the levels of contribution from Black students in the small groups. Over time, this work was taken into classroom settings, and a suite of "status interventions" were developed to promote equitable student participation [18]. This suite of strategies became known as *Complex Instruction*.

Complex Instruction has four primary components: 1) group-worthy tasks; 2) multiple-ability framing; 3) delegation of authority through group roles, and 4) assigning competence. The first three strategies refer to the types of tasks used in the classroom, how the tasks are framed as requiring multiple competencies, and how students are organized to work in groups. Broadly, these techniques create an environment in which students are better able to recognize the contributions of their peers and of themselves. These strategies have been described extensively elsewhere [19]. The fourth strategy, *assigning competence*, is the focus of this manuscript (described in the next section). While the first three components of Complex Instruction refer to how an instructor sets up the classroom community and organizes the environment, assigning competence a targeted strategy that an instructor uses within that context to remedy status imbalances that arise.

The use of Complex Instruction has been studied extensively over the past few decades [6, 7]. These reviews document numerous studies comprising hundreds of classrooms. These studies show how overall, this suite of strategies can reduce status imbalances across a classroom [18]. Similarly, these studies show how these strategies can enhance collaboration in the classroom, resulting in greater learning overall on standardized tests [6]. However, despite this wealth of research, there is less work that directly connects student status and status interventions to student participation across social marker identities, like race, gender, and disability. This is especially true for undergraduate STEM classrooms because Complex Instruction has been used primarily in K-12 settings.

Status often corresponds to social marker identities. For example, much of the early work focused on group interactions between Black and White students (e.g., Cohen & Roper, 1972). Similarly, other, more contemporary researchers have stated an explicitly anti-racist purpose for these pedagogies:

> "[Complex Instruction] is a powerful anti-racist pedagogy because it attends to dismantling hierarchies of competence that are steeped in race as a diffuse status characteristic. Let's break down a system that claims that young people with dark skin are not smart. We will rebuild our classroom communities with focus on their brilliance and interdependence among all students." (Lisa Jilk, personal communication, March 23, 2021; as quoted in Lotan, 2022)

Thus, even though the concept of status does not inherently attach to social markers, as educational theory and empirical research has made clear, status is often linked to social identities [e.g., 20–23]. We take this step one further using a methodological approach grounded in the EQUIP tool [9], as we elaborate in the *Equity Analytics* section.

## Assigning competence

Assigning competence is a way for STEM educators to use their power to 1) help students perceived as high-status recognize and appreciate the intellectual contributions of students perceived as lower-status classmates, 2) help students perceived as low-status appreciate their intellectual abilities, and 3) to call attention to useful problem solving strategies that one or more students have devised in working on a problem [24]. In this way, assigning competence can also be seen as a form of feedback to students that helps shape the nature of social interactions, because the instructor is signaling *what* is important in the classroom and *who* is making those important contributions. Through the intentional use of assigning competence, STEM educators can redefine what it means to be smart at STEM, disrupt inequities, and broaden opportunities for participation.

Assigning competence is an advanced instructional strategy so implementing it productively has a few prerequisites. First, instructors must have sufficient pedagogical content knowledge [25] to analyze the contributions of students (even partially correct ones) and recognize how they can contribute to the disciplinary activity of the classroom. Second, instructors must believe in the inherent brilliance of all students, regardless of their individual status or the status of their social marker group [cf. the concept of "Black brilliance"; 26]. Third, the instructor must be intentional about the use of assigning competence; it is not something that simply happens by chance. While monitoring the contributions of students, instructors need to capitalize on opportunities that arise to elevate the status of specific students perceived as low status that they are attending to. These instructor interventions modify the expectations that students have of each other and themselves.

In our work, we describe assigning competence as a three-step process, and with any component missing, the implementation is incomplete:

1. Identify a low status student.

2. Identify a disciplinary contribution from that student.

3. Make the contribution public in a way that positively highlights the contribution.

As the first step implies, an instructor cannot use assigning competence unless they have awareness of the status of their students. Traditionally, in Complex Instruction this would involve an instructor carefully observing students during group work time and making note of student status based on patterns of interaction. An instructor might also administer surveys to gather student perceptions of 1) competence and 2) popularity. In this paper, we offer another approach to identifying status, through the EQUIP tool [9], which we describe in the methods section.

The second step requires noticing a disciplinary contribution. This step is critical and differentiates the practice of assigning competence from empty praise. Praise is non-specific feedback such as "good job", "excellent", and "great work". Overall, research shows that these types of comments can reduce learning, because they focus the learner on themselves and public perceptions of them rather than on learning the content [27, 28]. Such comments become a problem when they do not provide students with clarity about what they did that was a "good job", "excellent", or warranted "great work." Instead, the teacher must recognize something that contributes to disciplinary activity, a specific concept, process, or way of thinking. In a math class, this would include using a representation to explain a key idea, connecting the ideas that their peers came up with, or explaining the context in which a mathematical idea is appropriate and when it is not. As such, if a student has not made a significant *mathematical* contribution, then it would not be possible to assign competence to that student at that time.

The third component speaks to the teacher's power in the classroom. When an instructor elevates a student contribution publicly, it sends important messages to the class. It explains to students both what is valued in the discipline and positions that student as a competent contributor to disciplinary activity. Given the positioning role of assigning competence, the practice will be most impactful when comments are made public in a larger space (e.g., whole-class discussion) rather than in the context of small group interactions [29].

With this final step, there are a variety of ways in which an instructor can make a student contribution public. Here we draw upon a framework that categorizes various forms of assigning competence [10]. We describe these categories in Table 1, providing examples from episodes in this paper.

## Equity analytics

To support the study of status and to track changes in student status, we utilized the *equity analytics* methodology [8, 9]. Equity analytics considers the distribution of resources—well-trained teachers, high-quality curriculum, opportunities to participate, and so forth—and how those could contribute to student success. In this work, we focus on one specific resource, classroom participation. The reason for this focus is three-fold:

1. participation promotes learning [30, 31];

2. participation supports identity development [32]

3. participation tends to be inequitable across social marker identities and contexts [23, e.g., 29, 33–35]

From the perspective of status, classroom participation gives us a window into student status. Prior research shows that students perceived as lower-status students tend to participate less, and higher-status students tend to participate more [7]. Thus, we see participation as an important indicator of status, and by tracking student participation, we can measure the impact of instructional interventions.

The use of equity analytics also supported the design of the professional development, through an *equity learning community* [8]. An equity learning community is a specific approach to faculty professional development, in which participants meet regularly to reflect on equity data (generated by the EQUIP tool; described further in the methods section), implement new teaching practices, and then collect data to study the impact of their changes. This professional learning is deep, ongoing, and iterative in a way that leads to concrete changes in practice. The use of this approach and its subsequent impact on improving equity in classroom participation has been widely documented [34, 36, 37].

**Table 1. Four categories of assigning competence.**

| Category | Description | Example |
|---|---|---|
| Highlighting, clarifying, and amplifying contributions | An instructor revoices, restates, or otherwise makes details of the student contribution explicit. | "The really creative thing that I liked about her approach is that they looked at the bedrooms. . .they used that approach to estimate the number of people [for water usage]." |
| Supporting specificity | An instructor follows up on a student idea to support them to make specific details of the contribution public. | "Okay, so [you used] the word row equivalent. . .so you think these are equal? What is B being row equivalent to A mean?" |
| Recognizing emergent ideas | An instructor validates an emergent or partially correct solution as valuable. | "That's not exactly correct, but concavity is relevant to the solution." |
| Validating unprompted attention to disciplinary ideas | Responding to students' unprompted contributions to make the details of their thinking explicit. | "Oh, you're right. 6.9 comes from the tank, so it's not going to the sewer. I was overthinking that. . .thanks for correcting me." |

To be clear, when it comes to issues of equity and status, it may be impossible to define an "ideal" distribution, especially as an outside observer. Nevertheless, we argue it is possible to define "inequitable." From our perspective, any attempt to produce equity and remedy status inequities must account for historical injustice [8]. At the very least, historically minoritized learners should have an equal share of resources allotted to them, but they may need more than an equal share to account for historical injustice. As such, we attend explicitly to issues of race, gender, disability, and other marginalized social marker identities in our determinations of status and its changes over time.

## Methods

### IRB approval

Human subjects research was conducted with approval from the Institutional Review Board at San Diego State University. All participating instructors provided written consent to be included in the study.

### Participants and context

This manuscript analyzes data from a larger, longitudinal project that focused on how equity analytics can promote equitable instruction in STEM [8]. To date, a total of 19 faculty members across the US have participated in five cohorts. These faculty members have come from a variety of disciplines (e.g., engineering, mathematics, public health). Across this larger sample, all instructors were focused on improving equity, but not all of them used assigning competence. Just two of the five cohorts discussed assigning competence as an instructional strategy, and from those two cohorts, only three faculty members participated in the program for an entire year. Accordingly, we focus on this subset of the faculty, just three—Anne, Ramesh, and Sam—who developed some facility with the strategy and used it on a somewhat regular basis. Our goal is not to claim that all instructors in the project used assigning competence in productive ways (although we believe they could be taught to), but rather, to illuminate what happens when this strategy is used effectively and with intentionality in undergraduate STEM classrooms. While each of these instructors participated in the project for multiple semesters, for each we chose a focal semester in which we captured data of them using assigning competence.

Anne was a White woman teaching mathematics at an elite private university. Her class consisted of 29 students with 11 women, 17 men, and one student with unspecified gender. The racial demographics were 14 White students, 11 Asian students, and 4 students with unknown race. Ramesh was an Asian American man teaching mathematics at a women's college. There were 18 cis-gender women in his class, 16 were white, 1 was Latinx, and 1 was Southeast Asian. Given the racial and gender homogeneity we looked at class as a sign of status. There was 1 first-generation college student, 13 students receiving financial aid, and 5 students who worked part-time. Sam was a white man teaching engineering at a Hispanic Serving Institution. Sam's class had 39 students, with 23 men, 2 nonbinary students, and 14 women. The racial demographics included 5 Asian students, 5 Latinx students, 1 Latinx/Asian student, 6 Middle Eastern Students, 16 White students, and 6 students of unknown race.

### Design

The three focal faculty members each participated in equity learning communities. Each participant began with an intake interview (conducted by the first author), to begin building relationships and to understand their equity goals and background. Each faculty participant

collected demographic data from their students using an electronic survey. Finally, faculty participants supported data collection by setting up a video camera in the front of their classroom (four lessons per semester), and the videos were shared with the student coder (who differed across learning communities) to be coded and for feedback to be generated. Both Anne and Ramesh were a part of the same learning community (with just two faculty, a student coder, and coach), while Sam participated in multiple communities over different semesters (both cross-disciplinary, and a group with three engineering faculty).

Learning communities met each month to debrief on data analytics and set intentional strategies for working with their students. Before the meetings, faculty received a feedback report that detailed the observation of their class, including trends in participation by students and social marker groups. Although the facilitators offered a variety of teaching strategies during the meetings, here we only focus on the assigning competence strategy.

## Data collection

We collected a variety of data for analysis. The primary data sources were generated as a part of the professional development process. These included recordings of the classroom observations, EQUIP analytics, feedback reports, recordings of the coaching debrief meetings, and so forth. We also recorded intake and exit interviews with each participant. These secondary data sources were used for triangulation purposes only. We also offer one vignette of assigning competence that was not captured in the classroom observations, because Sam spontaneously described it in his exit interview.

## Analytic approach

**Categorizing instances of assigning competence.** To identify instances of assigning competence, a team of three graduate students watched all the classroom observations (N = 12 total) multiple times to capture positive public interactions between the instructor and a student during whole class discussions. Our initial pass through the data was inclusive, and we included episodes that were solely praise, to ensure we didn't miss any instances of assigning competence. Next, the team worked together to discuss the episodes until actual uses of assigning competence were identified. There was a total of 26 instances of assigning competence identified across 20 students (Anne = 9; Ramesh = 10; Sam = 7). Accordingly, we observed approximately two instances of assigning competence per observed lesson, and we infer that had we recorded the entire semester of instruction there would be other instances that we missed. Thus, this analysis only provides us with a snapshot and a window into the larger process, as is the case with classroom observation-based methods.

To categorize these 26 instances we use an existing framework as described in Table 1 [10]. The coding process was completed by one of the graduate students, and a second graduate student coded 9 of the 26 episodes (34% of the dataset double coded) for interrater reliability. There was complete agreement on all 9 episodes that were double coded, for 100% agreement in coding.

**The EQUIP tool.** We used data generated from the EQUIP tool to track student participation [9]. The EQUIP data were initially generated as part of the professional development process to support faculty members to reflect on patterns of inequitable participation. We also used these same analytics to track student participation. The EQUIP observation tool was designed to answer questions about student participation broken down by social marker groups. For example, the tool can be used to answer questions like: 1) What percentage of contributions to whole-class discussions came from Black women in the class? Or, 2) What percentage of questions asked by the teacher were addressed towards men of color? EQUIP can

answer such questions because student behaviors (and associated teacher moves) are coded at the level of a student, and when demographic information is added (typically collected through a student survey), individual contributions can be aggregated up to group-level statistics. For the present paper, this also allowed us to capture the trajectories of individual students in the classes.

EQUIP focuses on the unit of analysis of *contribution*, which is continuous engagement from a single student not interrupted by another student. Defined in this way, a single contribution may include back-and-forth between a teacher and a student, which allows for EQUIP to be sensitive to teacher moves like teacher press that elevate the quality of a student contribution. Typically, EQUIP has been used to capture verbal talk. Similarly, in this manuscript, engagement was mostly verbal talk, but in some cases, we also captured student gestures that were made public. In forthcoming work from another project, we have customized EQUIP to record a wide variety of different modalities of participation.

After student contributions are coded, they are coded along any number of dimensions (e.g., length of student talk, type of teacher question), which provide a richer description of classroom engagement. For professional development purposes, contributions were coded along several dimensions and combined with student demographics to track broader patterns of equity. For the purposes of reporting in this manuscript, we focus solely on the number of contributions from individual students, because this allowed us to quantitively track their participations and status. For further information about other EQUIP dimensions, we point the reader to prior work [9, 23, 33].

Finally, once classroom observations are coded using the EQUIP protocol, the freely available EQUIP web app (https://www.equip.ninja) can be used to automatically generate data analytics for instructors to reflect upon. These analytics include representations of individual students (e.g., a heatmap or individual bar chart), representations of groups (e.g., a comparative bar chart of social marker groups), and timelines of changes over time. All these analytics were used to support the professional development process but were not used for our analyses below.

**Cluster analysis.** As indicators of status, we considered both student participation and demographics (if they were appropriate). We performed a cluster analysis to look for patterns in the quantity in student participation that would be indicative of different levels of status. We looked at relative levels of participation (rather than absolute levels of participation), given that some students participating *more* than others would indicate relative differences in status. Thus, the cluster analysis provided us with a succinct way to capture relative levels of participation. If a student moved into a higher cluster of participation, that would indicate that they were participating relatively more than the students in lower clusters.

We explored the use of both $k$-means and PAM cluster analysis and found that $k$-means was better able to capture variation in the data (whereas PAM flattened too many students into the same category). We used $k = 4$ clusters for the analysis. These clusters represented categories of low, middle, and high participation, with a fourth category to catch severe outliers, because $k$-means is sensitive to outliers. These outliers were then combined into the "high" category, so that we could have three levels of status. Next, we identified the relationship between assigning competence and changes in student participation. To do so, we tracked the trajectories of all students to see whether their participation increased (i.e., moving into a higher cluster), or if there was no increase (i.e., staying in the same cluster or moving to a lower cluster). Then, we computed an odds ratio of increasing participation based on whether the instructor had used assigning competence targeted towards that student.

**Focal student vignettes.** To provide a qualitative picture of the impact of assigning competence, we provide episodes from a few focal students. We used three selection criteria for the

focal students. First, there needed to be evidence that this was a student perceived as low status. We used a combination of EQUIP data showing low levels of participation and student demographic information showing they were minoritized in their discipline to select students. Second, we needed to have captured an instance of an instructor using assigning competence with this specific student. Third, we wanted to select students who had an upward trajectory in their participation. Finally, we reviewed the pool of potential focal students to identify students whose participation was adequately captured in richness throughout the video recordings to highlight what the impact looked like qualitatively. Although we cannot causally state that the use of assigning competence was the *only* factor improving participation, these analyses suggest that the use of assigning competence was related to the change. In many studies of Complex Instruction, the multiple-ability treatment (or framing) is used to set up the expectation that multiple competencies are needed to complete tasks, which then sets the stage for the effective use of assigning competence. In this study, instructors were not taught to use a multiple-ability framing, but we still found an impact of assigning competence used in isolation. Once we identified the focal students, we went through all the video records and transcribed every episode in which that student participated to generate vignettes.

## Results

### Use of assigning competence

We begin with an overview of the uses of assigning competence from faculty participants. These results are summarized in Table 2. As Table 2 shows, all four categorizations of assigning competence were present in our dataset, but the category of "highlighting, clarifying, and amplifying" was by far the most prevalent (65.4% of instances). We hypothesize that the prevalence of this form of assigning competence is likely due to the nature of our professional development, and the way that the project team framed assigning competence. Because we focused on identifying the contribution and then making it public, this is most closely aligned with the "highlighting, clarifying, and amplifying" category, which is why we suspect this is how faculty members most often used the practice.

### Impact on student participation

Our next set of analyses focused on the impact on student participation. Table 3 shows the number of students who had relatively increasing participation for those who were assigned competence and those who were not.

As Table 3 shows, some students relative level of participation increased regardless of the use of assigning competence. Simultaneously, students' relative levels of participation were much more likely to increase when instructors used assigning competence. The odds ratio of having a status increase for assigning competence compared to no assigning competence was 3.71. We found a significant result with Fisher's exact test ($p = 0.01839$). Thus, when instructors used assigning competence with specific students, those students were nearly four times as

**Table 2. Use of assigning competence.**

| Category | Anne | Ramesh | Sam | Total |
|---|---|---|---|---|
| Highlighting, clarifying, and amplifying | 6 | 6 | 5 | 17 |
| Supporting specificity | 3 | 2 | 1 | 6 |
| Recognizing emergent ideas | - | 1 | - | 1 |
| Validating unprompted attention | - | 1 | 1 | 2 |

**Table 3. Changes in student participation based on assigning competence.**

|  | Participation Increase | No Increase |
|---|---|---|
| Assigning Competence | 13 | 7 |
| No Assigning Competence | 22 | 44 |

likely as their peers who were not assigned competence to increase their participation (as evidenced by reaching higher clusters of relative participation) and this effect was significant.

In Fig 1, we provide a visualization of this increase for a single classroom, Ramesh's (Anne and Sam had similar visualizations, except that fewer students were assigned competence).

As Fig 1 shows, overall, the students who were assigned competence were more likely to increase in their relative participation. This was not uniform though; three students remained in the low cluster, and one student with higher participation (in the first observation) reduced to the middle cluster. Nonetheless, for four of eight students who were assigned competence, they had a positive trajectory of increased relative levels of participation (including the focal student Molly, described below). For students who were not assigned competence, the picture was very different (Fig 2). Only three students increased in relative participation (from low to

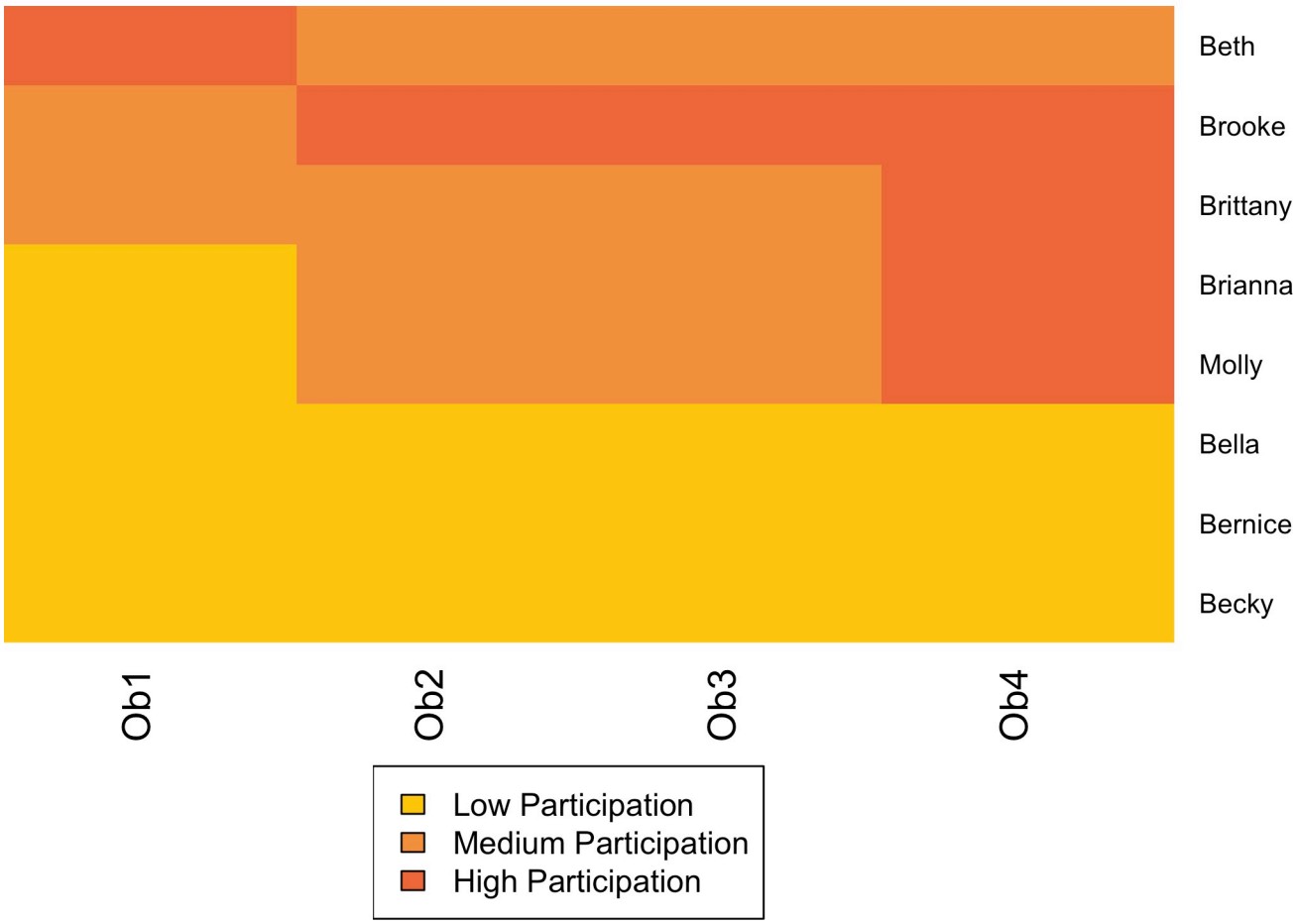

**Fig 1. Changes in student participation resulting from assigning competence in Ramesh's class.**

## Change in Student Participation (No Assigning Competence)

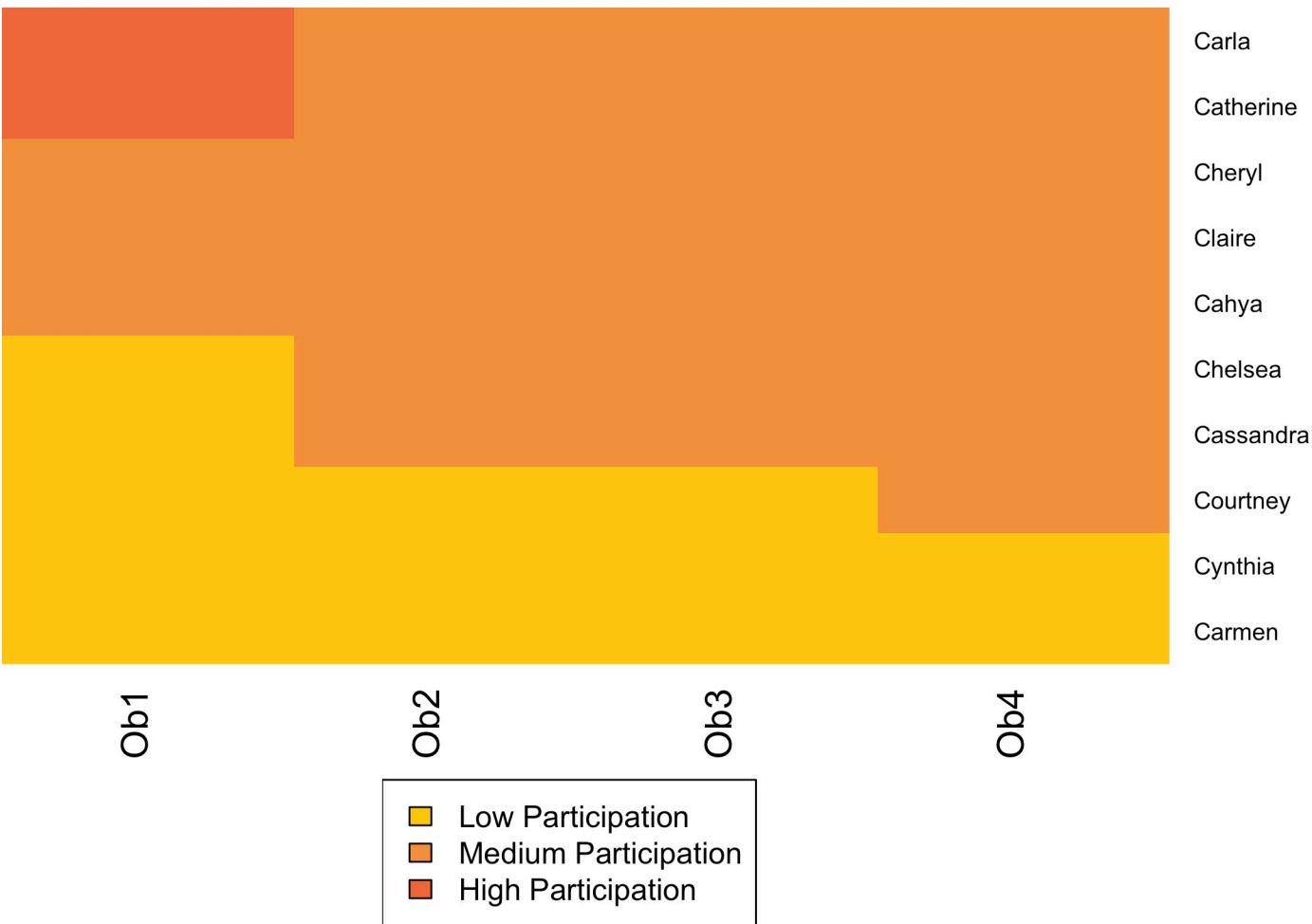

**Fig 2. Changes in student status not resulting from assigning competence in Ramesh's class.**

medium), and the rest decreased. Not a single student in the class who was not assigned competence reached the highest cluster of participation. Of course, these results are from a single classroom, but they underscore the potential power of this teaching technique.

We also looked at status as a factor in these changes. Notably, Beth was the only student who coded as high status based on high levels of participation in the first observation and high socioeconomic status. Students Bernice, Bella, Brianna, Brittany, Brooke, and Molly were all classified as low status based on low levels of prior participation and either receiving financial aid or having a job. Becky was considered moderately low status based on no prior participation but was of higher socioeconomic status. Of course, our participatory and demographic markers are only one way of operationalizing status, but overall, in this classroom, the positive impacts of assigning competence corresponded with students beginning with lower status. This shows that through intentional instructional moves, Ramesh was able to reshape status expectations in his classroom, which consequently changed relative levels of participation between students.

## Student vignettes

Having summarized the impact on student participation resulting from assigning competence, here we provide qualitative vignettes of participation to show the changing nature of participation from these students. Additionally, Fig 3 shows the changes in *total* participation over time. As Fig 3 shows, for the three focal students, they rarely contributed publicly before the intervention. For these students, only two contributions were witnessed over six observed lessons. While it is possible that the students participated across the lessons we had not observed, overall, this helped provide us with a baseline to understand the impact of assigning competence. In the sections that follow we trace the trajectories of each student.

**Case 1: Molly.** Molly was a white woman in Ramesh's class, who received financial aid. During the first observed class session, Molly never participated publicly. Her first instance of participation took place in a lighthearted social discussion related to an upcoming exam, during Observation 2. Later during that observation, Ramesh was discussing how to find the area under a curve using Riemann sums, and whether a given sum would be an overestimate or an underestimate of the true area. In this episode, Molly made a gesture with her hands that Ramesh utilized as an opportunity for assigning competence.

> Ramesh: Okay, is this an overestimate, an underestimate or are we not sure?
>
> [Choral Response: overestimate]
>
> Ramesh: Why is it an overestimate?
>
> Beth: Because you're taking area that's above the curve.
>
> Molly: [gestures area above the curve with slanted hands].
>
> Ramesh: So, Molly was doing things with her hands that are exactly correct, and Beth was explaining them at the same time.

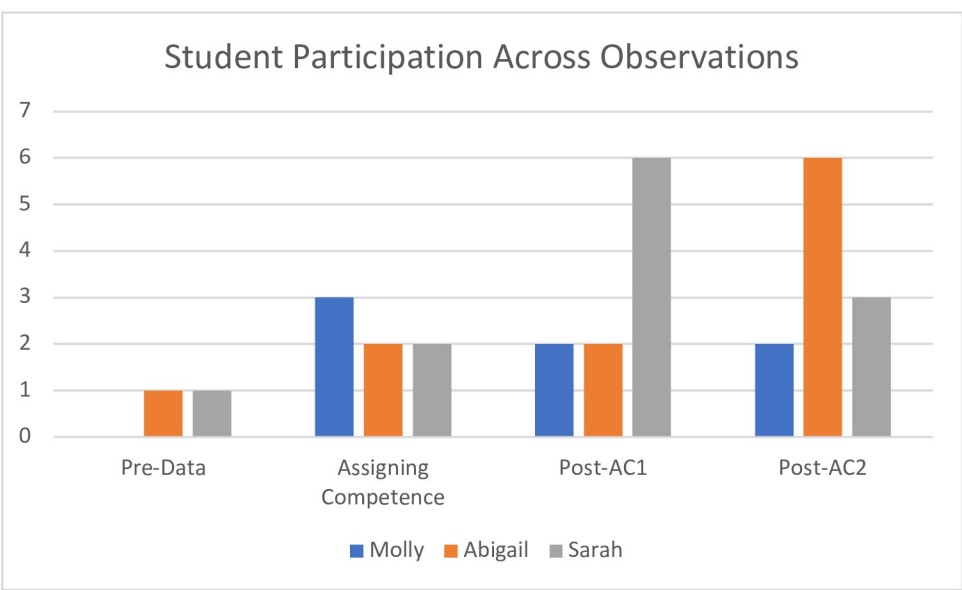

**Fig 3. Student participation trajectories before, during, and after assigning competence.** * Each of these time points consists of a single observation, except the "Pre-Data" category for Sarah, which has four observations (with only one contribution over all four observations).

[Ramesh continued working on the example.]

Ramesh: What has to be true about *f* for the right hand to be the overestimate and the left-hand to be the underestimate?

Molly and Brittany: [raise hands]

Ramesh: Brittany.

This is the episode of assigning competence through "highlighting, clarifying, and amplifying" a contribution. This episode is particularly notable, because Molly contributed to the discussion only through gesture, yet Ramesh recognized this as an important mathematical contribution, and made it public to the class. Ramesh identified her gesture as "exactly correct" which showcased her mathematical reasoning that otherwise would've been invisible to many students.

Later in the same lesson Molly participated verbally for the first time ever observed. Here, Ramesh asked the class what must be true about the function for them to not know whether the Riemann sum would be an over- or underestimate after Brittany shared her explanation of what was true about the function regarding right-hand sums.

Ramesh: I saw a couple other hands up, maybe someone else wants to address in what situations can I not say if these will be too big or too small. [iterates current observations]. What's true about the function when we aren't sure?

[Brooke raises hand]

Ramesh: Brooke.

Brooke: I'm not sure if this is the right answer, but does it have to do with the concavity?

[*Ramesh recognizes Brooke's emergent ideas*]

Ramesh: Not exactly. The concavity can be relevant, but we still know if it's increasing it doesn't matter, but–Molly?

Molly: I feel like if the function is constant then we should [be able to say].

Ramesh: Yeah, if it's constant then we'll have exact for both.

Ramesh: Maybe I'm not phrasing this question correctly–Brianna?

Brianna: If the function is a parabola, and stuff like that, the left hand will be over for the beginning and then an underestimate after that.

This episode was notable because Molly was comfortable making a verbal contribution even though she wasn't entirely sure of the answer. In response to her contribution, Ramesh asserted that she was correct. Although this wasn't coded as an instance of assigning competence, it still provided validation of Molly's ideas publicly.

Molly continued to participate across Observations 3 and 4. For example, in Observation 3, the class was discussing techniques for integrating improper integrals. Ramesh showed the class the graph of a function and asked them to discuss the improper integral of the function from negative infinity to positive infinity based on the visual area under the graph.

Ramesh: Limit from -infinity to infinity, what's gonna happen?

Molly: It's just 0.

Ramesh: Ah! Molly is thinking about the harder half of this. The harder half is "What is happening over here?" I don't know, who cares. But what's happening over here? [Points to part of the graph, but Ramesh's gestures are not captured by the camera.]

Again, Ramesh validates Molly's ideas by stating she is "thinking about the harder half" of the problem, implying that Molly was considering a more difficult question than the one Ramesh was posing. Even though Molly's idea in this episode isn't necessarily correct, he continues to provide encouragement for her to participate. This encouragement supported Molly's continued engagement, as evidenced in Observation 4. During this final episode, Ramesh was answering homework questions on the topic of sum and series convergence. Brittany had asked for help on a specific problem and Ramesh looked at which problem she was referring to. In the following exchange, Ramesh offered a suggestion to the problem, and Molly helped correct his feedback.

Ramesh: I'm going to let you all struggle with this one, but my hint is- right, we had done basically this problem, but we had done it with n times n plus 2, right? I guess ours was j's, this is n, but it doesn't matter what letter we use. But that's the method. Brittany, is that enough?

Brittany: Yes, did the question ask to find where it converged to? Because I-

Ramesh: Oh, yeah.

Molly: No, it just asked for the next one.

Ramesh: Oh, it just said find the first 5 terms.

Brianna: Yeah, I just checked my, the back of the book, and I-

Molly: Yeah, I got-

Brianna: I got 1 over 2 plus 1 over 6.

Molly: Yeah, and you just keep adding them, right.

Ramesh: Yeah, find the first five terms in the sequence of partial sums. So the first one is one-half plus one-sixth-

Brittany: Oh, I see! I hadn't added them.

This interaction is particularly notable because we see Molly having the confidence to correct a mistake from her instructor. Ramesh had mistakenly affirmed that the problem asked for where the series converged to, and Molly clarified that the problem did not ask for this, but only asked for the first five terms in the sequence of partial sums. Brianna validated Molly's feedback and the two students attempted to clarify how to find the partial sums. Looking back to see that Molly never participated in the initial observation, we can see that Ramesh's validation of her ideas and use of assigning competence had a notable impact.

**Case 2: Abigail.**   Abigail was an Asian American woman in Anne's class. In the first observation, Abigail only made one small contribution to the class discussion. When Anne asked the class to describe the meaning of a solution in "matrix language," Abigail responded that it was "the coefficients." Later in the same lesson, Anne asked the students how confident they were feeling about the material, and Abigail shook her head no (alongside another student). Anne publicly thanked both students for being transparent about their confidence levels. Overall, the first observation was primarily whole-class instruction, and dominated by men in

the class. Over time, Anne learned to break students into pairs as was suggested in her learning community.

During Observation 2, Anne used the strategy of pair discussions to create an opportunity for assigning competence. Anne split the students into pairs to work on a problem, and while students were working Anne monitored the discussions so that she could intentionally choose students to share.

Anne: I want to know: what is the relationship between these three spaces? The row space, null space, and column space of the matrix $A$ in a row-reduced form. Makes sense? [. . .]

Anne: Talk to your partner for a moment, and ask, are they the same? Are these two spaces the same? Are they related at all? If you have an answer, you've got to have a reason for that answer.

[students talk in pairs for about 6 minutes; during this time, Anne speaks with Abigail's group for about one minute, but the audio is not captured on camera.]

Anne: Let's come together and talk about these. These aren't really easy questions, right? That's great, because otherwise you wouldn't be learning anything. All right.

Anne: How about Abigail and Suparna, what did you decide about the row space?

Abigail: We decided that the row spaces are equivalent because A and its image are both equivalent.

Anne: So, the word row equivalent. . .you think these are equal? What does $B$ being row equivalent to $A$ mean?

Abigail: It means that any linear combination of $B$ also contains part of $A$, the row space of $A$.

Anne: Okay, I like that. So, the linear combinations of the rows of $A$ give you rows of $B$. And so, their span should be the same. So, row equivalent actually means we got from $A$ to $B$ using elementary row operations.

This episode of assigning competence was classified as "supporting specificity." Notably, Anne frames the task as challenging by describing the task as a "not really easy." When Abigail offers an idea, Anne probes deeper to have her explain what she means by "row equivalent." Afterwards, Anne responds, "I like that" and revoices Abigail's contribution to make it public for others. Overall, this episode highlights how Anne created a context within which she could assign competence to a woman in the class (which was an instructional goal of hers). First, by breaking up students into pairs, she created an opportunity to listen for the contributions. Next, she helped Abigail make her contribution public and positioned it positively in front of the class.

The impact of Anne's strategies became more visible over time. In Observation 3, women in the class began to share their ideas with more confidence and regularity. For example, there was a lively discussion about orthogonality, and in this context, Abigail felt comfortable sharing her ideas without being called on.

Anne: "I want to start with talking about orthogonality. Does anyone know what we mean by [orthogonality]?"

Abigail: [not called on] Perpendicular!

Anne: "Okay, that's the symbol for perpendicular. How can we tell whether two vectors are perpendicular?"

Anu: [raises hand] "The dot product."

Anne: "The dot product. What do you do with the dot product?"

Anu: "You see if it's equal to zero?"

Anne: "Okay. Nancy, what were you going to say?"

Nancy: "The same thing. The dot product is equal to zero."

Here we see three women contributing ideas confidently. Later in Observation 3, we observed Anne further assigning competence to Abigail's work.

Anne: "Okay, why don't you work on this problem too? [Finding the basis for the image of T under some linear transformation.]"

Students: [working in groups for 12 minutes while Anne checks in with each group]

Anne: "Let's just go over this second problem. I really like what I heard over at Abigail and Nico's table, because they used language that we've been using in class."

Anne: "So, the image of T, is the set of, it lives in R-Four, so it's gonna be the set {a, b, c, d} [. . .]"

Anne: A lot of you said, okay, the basis, so this looks like, whatever a is, c is the negative of that. [Anne continues working out the problem with participation from students.]

Anne: "What I heard from Abigail's table was that, well, we need to use the expansion theorem to say, 'well, what do I need to add to this to get a basis for R4 [. . .]'"

Anne: "So what did you get, Abigail?

Abigail: "I used the [. . .]"

Anne: [Writing out computations on the board]

Anne: I think a lot of you nailed that puppy! Yes! A good place to start our weekend! [students applaud.]

We coded this example as "highlighting, clarifying, and amplifying." This example has a few key features. Again, Anne broke students into pairs to monitor their contributions. Twice in the discussion Anne draws attention to Abigail's contribution, highlighting a clear mathematical contribution of using "language that we've been using in class." Later, Anne enthusiastically exclaims "You nailed that puppy!" highlighting her enthusiasm for the mathematical contribution and her ability to draw on specific disciplinary concepts that the class was discussing.

Given the repeated acknowledgement and elevation of her comments from Anne, not surprisingly, Abigail continued to demonstrate a higher level of comfort when participating in the classroom during Observation 4. This was evidenced by her shouting out answers to questions without being called on, as in the following episode.

Anne: [writes the question on the board] Okay, why can I write any vector in R2 as a linear combination of those two vectors?

Abigail: [not called on] Oh, because b is the basis?

Anne: Because [. . .]

Abigail: Everything in R2 can be expressed as a linear combination of the elements of b.

Anne: Cool. Okay so [completes computations]

After Abigail offers her explanation, Anne acknowledges it with "cool." This type of acknowledgement is not an example of assigning competence but shows that Anne continued to validate the ideas of her students, specifically Abigail in this case.

**Case 3: Sarah.**  Sarah was a woman of unknown race in Sam's class. Of all the focal students, Sarah had the longest baseline data: across four classroom observations she only participated a single time, and it was a relatively low-level form of contribution. The class was solving problems using Excel, and Sarah asked a clarifying question, "Do you have to highlight the whole table or like say you have a ton of rows?" She didn't offer any of her own disciplinary ideas, yet.

During Observation 5, Sam tried a new instructional strategy in his class that was conducted over Zoom. This time, he asked all the students to share their screens as a part of a whole-class discussion so that students could see each other's work. After displaying her plots publicly, Sam commented on the quality of Sarah's work.

Sam: Alright, Sarah your plots look really good, and I see you're starting on the pumping rate and the cumulative tank volume. So, I see that you put a standard pumping rate of 2000. That's a really good number, how did you arrive at that number? This question is for Sarah.

Sarah: I looked at the hourly consumption and I just kinda guessed at like a good average.

Sam: Awesome, yep, that's exactly what you wanna do.

We coded this as an instance of "supporting specificity" for Sarah's contribution. Sam set up a situation in which all students would have an opportunity to participate and looked at her plots to see the standard pumping rate. After stating it was a "really good number" he pressed Sarah to share details of her thinking. Later in the discussion, Sarah asks a question about Sam's calculations of average water consumption, which leads Sam to suggest for all students to add a column to their calculations.

Sarah: Is the column for F, like, the one after the pumping rate supposed to be the volume of the tank?

Sam: Yup, and you can add, actually, I'll encourage everyone to add in between E and F, if you can add a column in there, right click on F, up on the top and then you go to insert and then add a column in between that is pumping rate gallons. [. . .]

Although this wasn't considered assigning competence, we see that Sam is validating Sarah's question as valuable by suggesting the rest of the class to attend to it.

In Observation 6, we see the impact of Sam's validation of Sarah's ideas. When Sam was working out a problem, Sarah has the confidence to publicly correct his mistake.

Sam: Drainage water from the hand washing station [. . .] supplement that with potable water so 8.2 minus 6.9 you get that response, multiply that by 5 that's how much water savings this device right here would cause, would result in, for each household of five people.

Student: 6.5

Sarah: Why would you subtract it? Wouldn't the saved water be just the 6.9 times 5?

Sam: You'd be saving water from what usually fills up the tank back here. So usually, potable water is used to fill up this tank. Right, on average that's 8.2 gallons per person per day. So instead of using 8.2 gallons of potable water to fill up this tank, you're using 6.9 of those gallons are coming from the sink.

Sarah: Right, so if you're asking how much water would be saved, wouldn't it be 6.9 times 5, because if you subtract it that's how much water you still need of potable water, right?

Sam: Oh yeah, you are, you're right [. . .] Yes, because you save 6.9, let's think about this, 6.9 comes into the tank, so it's not going to the sewer, so it is 6.9 times 5. Very simple, ok. I was overthinking that and I forgot that it was that simple. Thank you for correcting me. Who is it that corrected me by the way? I'll give you a thousand extra points.

Rochelle: I think it was Sarah.

Sam: Awesome, Sarah, I'm gonna give you a thousand extra credit points. Because I do want you to correct me when you see me say something wrong. You know, I'm a person and I make mistakes sometimes and miscalculate things sometimes.

Here, we see Sarah asks a question to clarify one of Sam's computations. Sam initially brushes off her response with a quick explanation, yet Sarah persists in asking again, explaining to Sam where the error was. This change in confidence from Sarah is notable given she only participated once in the first four lessons! It's notable that a peer, Rochelle, notices Sarah's competence and recognizes her contribution. Then, Sam validates her correction by giving her "a thousand extra credit points." We coded this as "validating unprompted attention to disciplinary ideas." In his exit interview, Sam brought up this experience. He recalled being both "embarrassed" and "proud" of Sarah for interrupting him, recalling that she was a student who rarely if ever participated at the beginning of the semester.

The strategy of assigning competence was particularly salient for Sam and is one that he talked about extensively in his exit interview. He also described a situation in an earlier semester in which there was a group of students working together in a breakout room. There was a white man in the group who spoke frequently in class, who was convinced that he was correct in solving a problem. Marina, a woman of color in the same group, tried to correct his incorrect solution, but he continued to argue with her. When Sam entered the breakout room, he noticed that Marina was correct, and he highlighted her contribution within the small group. Later, he highlighted that contribution publicly to the whole class. This interaction was the first time he ever met Marina. After that semester, she got involved in one of Sam's research projects, and even served as a student representative for the department. Later, she continued to join the graduate engineering program and received a Master's degree. While these experiences cannot all causally be attributed to Sam's use of assigning competence, this set of interactions stood out for him as particularly notable.

## Discussion

This manuscript makes an important contribution to the research literature through the study of *assigning competence* in undergraduate STEM education. Although Complex Instruction has been studied extensively in K-12 settings, there has been little work in undergraduate STEM. Moreover, we are not aware of any work to date that has tracked the specific trajectories of student participation due to assigning competence. Thus, our connection of assigning competence to the EQUIP observation tool to capture changes in student participation provides a powerful methodology that equity scholars in our field can apply to other settings. This allows researchers to add elements of quantitative research to complement techniques that have already been studied more extensively through qualitative research, supporting mixed methods work. Moreover, from a teaching perspective, our focus on assigning competence shows that instructors can intentionally shift the participation dynamics of their classrooms through intentional instructional moves. Here we highlight a few key findings.

First, we found that instructors were capable of learning to use assigning competence. This is notable, because assigning competence is an advanced instructional strategy, and most of the participating instructors had received minimal prior professional development, especially around equitable teaching. Assigning competence can only be used when a low-status student makes an important disciplinary contribution (e.g., in math), the instructor notices that contribution, and makes it public. This entire setup takes a great deal of skill to orchestrate. Nevertheless, the instructors in this study learned to use the technique and managed to use it about twice per class session that we observed, on average. Using a classification scheme of types of assigning competence, we found that instructor attempts were primarily in the category of "highlighting, clarifying, and amplifying," which were 17/26 or 65.4% of instances. We suspect that the prevalence of this category may have been related to the nature of the professional development that instructors received, but we have limited data to make causal claims about this.

Second, the assigning competence strategies seemed to be effective in shaping student participation (which alongside student demographics, we used as an indicator of status). We found an odds ratio of 3.71 for increasing student relative participation over the course of the semester for students who were assigned competence compared to those who were not. This amounts to almost a fourfold increase in likelihood of increasing participation for students who were assigned competence! As we illustrated using Ramesh's class as an example (see Fig 1), many students who were assigned competence had very low levels of participation at the beginning of the semester, which increased in measurable ways. There were similar findings in other classrooms.

Third, this manuscript provides rich qualitative vignettes of three focal students, showing how assigning competence shaped not only the quantity of participation, but the nature of it. The focal students had minimal levels of participation before the use of assigning competence, and through the intentional validation of student ideas, instructors elevated them to become crucial voices in the classroom discussions. In the cases of both Molly and Sarah, they publicly corrected their instructor's mistakes, which demonstrates a high level of self-efficacy and confidence. The vignettes are also valuable because they provide a glimpse into how instructors created classroom situations in which they could effectively assign competence and increase the status of their students. For example, Anne used partner breakouts to monitor student contributions, and Sam created an "everyone shares" type of scenario where students could show graphs on their computer screens.

Finally, we reflect on the affordances of our approach for future research. Equity scholars have long argued that a "for all" approach to promoting equity is ineffective, as has been shown through empirical data [21, 38]. However, while the field of STEM education has many

classroom observation tools for generating data to study classroom practice, very few tools disaggregate by student social markers or by individual students, thus limiting their ability to study improvements to equity in classrooms. Through our work here, we show how the disaggregated features of the EQUIP tool can be used in conjunction with long-established equity techniques to provide new insights into their implementation and impact. We argue that this is a very productive line of inquiry for scholars in our field, and this manuscript provides one such model of how to use this approach.

## Limitations

There are a few important limitations to this work. The first is that we had a relatively modest sample size of 84 students, spread across 3 classrooms. Nonetheless, given the heterogeneity in classroom contexts and instructors, we suspect that we would have similar findings in other settings. Additionally, establishing causality can be a challenge in this type of research. Although we clearly demonstrate that participation increased for students after the use of assigning competence (and with an odds ratio of 3.71 compared to students who were not assigned competence), classrooms are complex spaces, and we cannot rule out the interaction of other factors both in terms of faculty instruction and student interactions. Follow up research to replicate these findings is recommended.

## Conclusion and implications

Promoting equitable instruction is an elusive imperative. Despite an increasing focus on this goal, there is still limited evidence for many instructional approaches to improving this equity. In contrast, this manuscript provides strong evidence of the impact of assigning competence to improve the classroom participation (and thus learning) of students who are historically marginalized in their disciplines. As such, we strongly recommend providing professional development to STEM faculty to learn assigning competence and other related strategies, so that they can improve equity in their classrooms in measurable ways. This manuscript provides a concrete instantiation of professional development with illustrative examples to help others take up this same work.

## Supporting information

**S1 Data.**
(CSV)

**S2 Data.**
(CSV)

**S3 Data.**
(CSV)

**S1 File.**
(PDF)

## Acknowledgments

We thank Rachel Lotan for her generative feedback on an earlier version of this manuscript.

## Author Contributions

**Conceptualization:** Daniel Lee Reinholz, Charles Wilkes II, Niral Shah.

**Data curation:** Daniel Lee Reinholz, Mariah Gabriella Moschetti.

**Formal analysis:** Daniel Lee Reinholz, Mariah Gabriella Moschetti, Jan Tracy Camacho, Eva Fuentes-Lopez.

**Funding acquisition:** Daniel Lee Reinholz.

**Investigation:** Daniel Lee Reinholz, Mariah Gabriella Moschetti.

**Methodology:** Daniel Lee Reinholz.

**Project administration:** Daniel Lee Reinholz, Niral Shah.

**Resources:** Daniel Lee Reinholz.

**Software:** Daniel Lee Reinholz, Niral Shah.

**Supervision:** Daniel Lee Reinholz.

**Validation:** Daniel Lee Reinholz.

**Visualization:** Daniel Lee Reinholz.

**Writing – original draft:** Daniel Lee Reinholz.

**Writing – review & editing:** Daniel Lee Reinholz, Mariah Gabriella Moschetti, Jan Tracy Camacho, Eva Fuentes-Lopez, Charles Wilkes II.

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
