## [Decision Letter · Decision Letter 0]

9 Oct 2023

PONE-D-23-23525Equity in practice: Assigning competence to shape STEM student participationPLOS ONE

Dear Dr. Reinholz,

Thank you for submitting your manuscript to PLOS ONE. After careful consideration, we feel that it has merit but does not fully meet PLOS ONE’s publication criteria as it currently stands. Therefore, we invite you to submit a revised version of the manuscript that addresses the points raised during the review process.

We look forward to receiving your revised manuscript.

Kind regards,

Jin Su Jeong, Ph.D.

Academic Editor

PLOS ONE

Reviewers' comments:

Reviewer's Responses to Questions

**Comments to the Author**

1. Is the manuscript technically sound, and do the data support the conclusions?

Reviewer #1: Yes

Reviewer #2: Yes

2. Has the statistical analysis been performed appropriately and rigorously? 

Reviewer #1: No

Reviewer #2: Yes

3. Have the authors made all data underlying the findings in their manuscript fully available?

Reviewer #1: No

Reviewer #2: No

4. Is the manuscript presented in an intelligible fashion and written in standard English?

Reviewer #1: Yes

Reviewer #2: Yes

5. Review Comments to the Author

Reviewer #1: I have had the opportunity to review your manuscript, and I must express my appreciation for the effort you have put into it. Your work is certainly promising, but there are a few areas that I believe require clarification and enhancement in order to meet the necessary standards for publication.

Firstly, in the introduction section, you mentioned a reference to a previously published study on line 55. I would like to understand why this study wasn't published in its entirety. Providing some context or rationale for this decision would be beneficial.

Secondly, the process of participant selection is not adequately elucidated in your manuscript. It's crucial to detail how the participants were chosen, especially as you focused on three specific faculties. Clarifying the rationale behind this selection and whether data collected from other participants exhibited similar trends would be valuable for readers.

In the methodology section, a more comprehensive explanation of the EQUIP analytics is required. The current description lacks clarity, making it difficult for readers to grasp the methodology's nuances.

Regarding the focal student vignettes, I believe further information on the selection criteria is needed. You mentioned the use of a selection criterion, but it would be beneficial to elaborate on this point. Additionally, I wonder if you explored other factors beyond "assigning competence" that influenced participation. For instance, were there any notable differences among instructors or the educational backgrounds of participants, particularly in terms of their mathematics knowledge? Given the consistent reference to math content in your study, it's worth discussing the choice of "STEM" in the title.

I recommend enhancing the discussion and conclusion sections to provide more depth and insight. A more profound exploration of the implications of your results for other researchers would be valuable to the academic community.

Lastly, in terms of limitations, I suggest considering the inclusion of data from a broader range of participants beyond the three faculties you focused on. Expanding the dataset could strengthen the robustness of your findings.

Reviewer #2: This paper seeks to provide empirical evidence for equitable teaching techniques. The research is grounded in Complex Instruction. The focus of the paper is on the application of assignment competence to undergraduate STEM classrooms. The claim is made that this is an original contribution documenting changes in student participation resulting from assigning competence in undergraduate STEM.

Research questions driving this exploration were clear and appropriate:

How did instructors leverage assigning competence as an instructional strategy to mitigate racial and gender inequities in their classroom participation?

How did the use of assigning competence impact classroom participation patterns?

Rationale and background

An argument is made for improving equity in STEM education within USA, particularly at tertiary level. The authors claim that the goals of equity and inclusion are nebulous and only seen through outcome data-too late for actions to be taken. This becomes a barrier to uptake and study of equitable teaching strategies. Limited empirical evidence documenting the impact of most inclusive teaching strategies-the exception being Complex Instruction. Less work on individual students who are perceived as low-status and from marginalised groups. This is the gap filled by the current work. This section is sound and clearly justifies the need for the current research.

Literature

Overview of Complex Instruction and prior studies of assigning competence are discussed.

Methodology

Data for this manuscript were drawn from a larger study that engaged university STEM faculty in sustained PLD. Data analytics describing patterns of student participation in their classrooms were used.

Student participation was tracked using the EQUIP observation tool which offered a rigorous methodological approach to understanding and tracking changes in student participation-an indicator of status. Equip focused on the unit of analysis of contribution-as continuous engagement from a single student not interrupted by another students. These were mostly verbal but some student gestures.

A variety of data analysed. Primary data were drawn from the PLD process-recordings of classroom observations, EQUIP analytics, feedback reports, recordings of coaching debrief meetings.

The coding process rigorous completed by one student and a second student double coded for interrater reliability

The exploration of using both k-means and PAM cluster analysis was clearly outlined, described, and justified as appropriate to catch categories of participation and outliers.

The rigor of the focal student vignettes was appropriate-these were clearly described, discussed, and justified. These highlighted the technical systems and methodology suitable for the purposes of this journal.

Results

An overview of instructors’ use of assigning competence and code all instance of assigning competence according to a typology of the strategy. Results were clearly outlined and linked to the systems of methodology. Discussion and conclusions highlighted both contributions, limitations, and foci for potential future research. Conclusions were presented in an appropriate manner and are supported by the data.

Overall, this original research has made reference to technical detail and rigor of collection, analysis, and discussion and meets the standards of research integrity.

6. PLOS authors have the option to publish the peer review history of their article (what does this mean?). If published, this will include your full peer review and any attached files.

Reviewer #1: No

Reviewer #2: No

---

## [Author Response · Author response to Decision Letter 0]

8 Dec 2023

To whom it may concern,

We appreciate the thoughtful response to our article. We are encouraged to hear that the reviewers find this article to be of interest to PLOS ONE, and we have revised the manuscript in accordance with their suggestions. Please see our line-by-line response below. Thank you for your time and consideration.

Kind regards,

-Authors

Reviewer #1: I have had the opportunity to review your manuscript, and I must express my appreciation for the effort you have put into it. Your work is certainly promising, but there are a few areas that I believe require clarification and enhancement in order to meet the necessary standards for publication.

Thank you for your encouragement and helpful feedback.

Firstly, in the introduction section, you mentioned a reference to a previously published study on line 55. I would like to understand why this study wasn't published in its entirety. Providing some context or rationale for this decision would be beneficial.

The overall study is a larger, $1 million study founded by the NSF over five years. The study was proposed to occur over multiple phases. Given the size and scope of the larger change project, it is not possible to report on the results in a single publication (not even the length of a book). As such, we have written smaller papers in order to focus on specific aspects of the large project in sufficient detail. We have added language to clarify the nature of the larger study and why this paper focuses on a subset of it.

Secondly, the process of participant selection is not adequately elucidated in your manuscript. It's crucial to detail how the participants were chosen, especially as you focused on three specific faculties. Clarifying the rationale behind this selection and whether data collected from other participants exhibited similar trends would be valuable for readers.

We have added additional information about participant selection in the methods section. In particular, the 19 faculty members participated across five cohorts, and just two of five cohorts discussed assigning competence Within those two cohorts, there were only three faculty members who participated for an entire year, and those faculty members are the focus of this paper. Given that each coach was able to focus on their own teaching strategies, we would not expect that all instructors across all cohorts would be taught to use assigning competence. In future iterations of the work, we may explicitly include this strategy for all participants, and then we would have a larger cohort to study. While faculty members in other cohorts did adopt strategies to improve equity, assigning competence was not one of the strategies they used. 

In the methodology section, a more comprehensive explanation of the EQUIP analytics is required. The current description lacks clarity, making it difficult for readers to grasp the methodology's nuances.

We have added extended detail to the EQUIP tool and the types of analytics it generates in the methods section.

Regarding the focal student vignettes, I believe further information on the selection criteria is needed. You mentioned the use of a selection criterion, but it would be beneficial to elaborate on this point. 

We have added additional detail explaining our choice of those particular students. 

Additionally, I wonder if you explored other factors beyond "assigning competence" that influenced participation. For instance, were there any notable differences among instructors or the educational backgrounds of participants, particularly in terms of their mathematics knowledge? Given the consistent reference to math content in your study, it's worth discussing the choice of "STEM" in the title.

We do not have surveys of students’ prior knowledge, but all students were placed into the same college courses so presumably there was some comparability in their background knowledge. Notably, the students who were initially low-status but were not assigned competence did not show the same upward trajectory as those who were.

For us, the importance of the math comes from the fact that assigning competence only works when the instructor acknowledged a mathematically significant contribution (or engineering in one case). This point is one that we have strengthened in our definition of assigning competence and in our discussion.

I recommend enhancing the discussion and conclusion sections to provide more depth and insight. A more profound exploration of the implications of your results for other researchers would be valuable to the academic community.

We have enhanced the discussion section to directly address the impact of this work on our research community.

Lastly, in terms of limitations, I suggest considering the inclusion of data from a broader range of participants beyond the three faculties you focused on. Expanding the dataset could strengthen the robustness of your findings.

We concur that this work would be strengthened with the inclusion of more faculty members. In the existing dataset, there are only three participants who used assigning competence over the long-term. However, this is an important focal area that we will consider for the work as it moves forward. Thanks again!

 

Reviewer #2: This paper seeks to provide empirical evidence for equitable teaching techniques. The research is grounded in Complex Instruction. The focus of the paper is on the application of assignment competence to undergraduate STEM classrooms. The claim is made that this is an original contribution documenting changes in student participation resulting from assigning competence in undergraduate STEM.

Thank you very much for your engagement with our paper and for your supportive feedback!

Research questions driving this exploration were clear and appropriate:

How did instructors leverage assigning competence as an instructional strategy to mitigate racial and gender inequities in their classroom participation?

How did the use of assigning competence impact classroom participation patterns?

Rationale and background

An argument is made for improving equity in STEM education within USA, particularly at tertiary level. The authors claim that the goals of equity and inclusion are nebulous and only seen through outcome data-too late for actions to be taken. This becomes a barrier to uptake and study of equitable teaching strategies. Limited empirical evidence documenting the impact of most inclusive teaching strategies-the exception being Complex Instruction. Less work on individual students who are perceived as low-status and from marginalised groups. This is the gap filled by the current work. This section is sound and clearly justifies the need for the current research.

Literature

Overview of Complex Instruction and prior studies of assigning competence are discussed.

Methodology

Data for this manuscript were drawn from a larger study that engaged university STEM faculty in sustained PLD. Data analytics describing patterns of student participation in their classrooms were used.

Student participation was tracked using the EQUIP observation tool which offered a rigorous methodological approach to understanding and tracking changes in student participation-an indicator of status. Equip focused on the unit of analysis of contribution-as continuous engagement from a single student not interrupted by another students. These were mostly verbal but some student gestures.

A variety of data analysed. Primary data were drawn from the PLD process-recordings of classroom observations, EQUIP analytics, feedback reports, recordings of coaching debrief meetings.

The coding process rigorous completed by one student and a second student double coded for interrater reliability

The exploration of using both k-means and PAM cluster analysis was clearly outlined, described, and justified as appropriate to catch categories of participation and outliers.

The rigor of the focal student vignettes was appropriate-these were clearly described, discussed, and justified. These highlighted the technical systems and methodology suitable for the purposes of this journal.

Results

An overview of instructors’ use of assigning competence and code all instance of assigning competence according to a typology of the strategy. Results were clearly outlined and linked to the systems of methodology. Discussion and conclusions highlighted both contributions, limitations, and foci for potential future research. Conclusions were presented in an appropriate manner and are supported by the data.

Overall, this original research has made reference to technical detail and rigor of collection, analysis, and discussion and meets the standards of research integrity.

---

## [Decision Letter · Decision Letter 1]

20 Feb 2024

Equity in practice: Assigning competence to shape STEM student participation

PONE-D-23-23525R1

Dear Dr. Reinholz,

We’re pleased to inform you that your manuscript has been judged scientifically suitable for publication and will be formally accepted for publication once it meets all outstanding technical requirements.

Kind regards,

Jin Su Jeong, Ph.D.

Academic Editor

PLOS ONE

Additional Editor Comments (optional):

Reviewers' comments:

Reviewer's Responses to Questions

**Comments to the Author**

1. If the authors have adequately addressed your comments raised in a previous round of review and you feel that this manuscript is now acceptable for publication, you may indicate that here to bypass the “Comments to the Author” section, enter your conflict of interest statement in the “Confidential to Editor” section, and submit your "Accept" recommendation.

Reviewer #1: All comments have been addressed

Reviewer #2: All comments have been addressed

2. Is the manuscript technically sound, and do the data support the conclusions?

Reviewer #1: Yes

Reviewer #2: Yes

3. Has the statistical analysis been performed appropriately and rigorously? 

Reviewer #1: Yes

Reviewer #2: Yes

4. Have the authors made all data underlying the findings in their manuscript fully available?

Reviewer #1: Yes

Reviewer #2: Yes

5. Is the manuscript presented in an intelligible fashion and written in standard English?

Reviewer #1: Yes

Reviewer #2: Yes

6. Review Comments to the Author

Reviewer #1: Dear Authors: I have revised the revised version of your MS and all my concerns have been answered/clarified. I appreciate the effort taken to prepare this revised version. I have recommended the editor to accept its publication. Regards

Reviewer #2: (No Response)

7. PLOS authors have the option to publish the peer review history of their article (what does this mean?). If published, this will include your full peer review and any attached files.

Reviewer #1: **Yes: **David González-Gómez

Reviewer #2: No

---

## [Editor Report · Acceptance letter]

4 Apr 2024

PONE-D-23-23525R1 

PLOS ONE

Dear Dr. Reinholz, 

I'm pleased to inform you that your manuscript has been deemed suitable for publication in PLOS ONE. Congratulations! Your manuscript is now being handed over to our production team.

Kind regards, 

on behalf of

Dr. Jin Su Jeong 

Academic Editor

PLOS ONE